# LncRNAs Act as a Link between Chronic Liver Disease and Hepatocellular Carcinoma

**DOI:** 10.3390/ijms21082883

**Published:** 2020-04-20

**Authors:** Young-Ah Kim, Kwan-Kyu Park, Sun-Jae Lee

**Affiliations:** Department of Pathology, Catholic University of Daegu School of Medicine, Daegu 42472, Korea; youngah7840@naver.com (Y.-A.K.); kkpark@cu.ac.kr (K.-K.P.)

**Keywords:** long non-coding RNAs, fibrosis, chronic liver disease, hepatocellular carcinoma

## Abstract

Long non-coding RNAs (lncRNAs) are emerging as important contributors to the biological processes underlying the pathophysiology of various human diseases, including hepatocellular carcinoma (HCC). However, the involvement of these molecules in chronic liver diseases, such as nonalcoholic fatty liver disease (NAFLD) and viral hepatitis, has only recently been considered in scientific research. While extensive studies on the pathogenesis of the development of HCC from hepatic fibrosis have been conducted, their regulatory molecular mechanisms are still only partially understood. The underlying mechanisms related to lncRNAs leading to HCC from chronic liver diseases and cirrhosis have not yet been entirely elucidated. Therefore, elucidating the functional roles of lncRNAs in chronic liver disease and HCC can contribute to a better understanding of the molecular mechanisms, and may help in developing novel diagnostic biomarkers and therapeutic targets for HCC, as well as in preventing the progression of chronic liver disease to HCC. Here, we comprehensively review and briefly summarize some lncRNAs that participate in both hepatic fibrosis and HCC.

## 1. Introduction

Long non-coding RNAs (lncRNAs) are a subgroup of non-coding RNA (ncRNAs) transcripts, which are greater than 200 nucleotides in length, with little or no protein-coding potential [1,2]. While lncRNAs are emerging as important contributors to the biological processes underlying the pathophysiology of various human diseases, including inflammation and neoplasia [3], the involvement of these molecules in chronic liver diseases, such as viral hepatitis and nonalcoholic fatty liver disease (NAFLD), has only been considered in scientific research [3].

Hepatocellular carcinoma (HCC) is one of the most common malignant tumors, with a poor prognosis and low survival rates [4,5]. Despite advances in the understanding of the molecular mechanisms of HCC, the overall low survival time, high rates of metastasis, postsurgical recurrence, and chemoresistance are still unsolved problems [5,6,7,8]. Hepatic fibrosis is a major risk factor for HCC, and it is a continuous wound-healing process that leads to the dysregulation of extracellular matrix (ECM) proteins and the distortion of normal liver architecture [9]. Many chronic liver diseases, such as viral hepatitis, alcohol toxicity, drug abuse, metabolic syndrome, and autoimmune hepatitis result in hepatic fibrosis or cirrhosis [10], which is the primary stage of HCC [11]. While extensive studies on the pathogenesis of the development of HCC from hepatic fibrosis have been performed, their regulatory molecular mechanisms are still only partially understood [11].

The recent application of next-generation sequencing (NGS), particularly RNA-sequencing (RNA-Seq), has improved our knowledge of lncRNAs related to various types of diseases [2]. With the application of NGS and high-resolution microarray techniques, many studies suggest that HCC-related lncRNAs could influence the initiation, progression and suppression of HCC [5]. However, the underlying mechanisms related to lncRNAs leading to HCC from chronic liver diseases and cirrhosis have not been entirely elucidated yet [12]. Therefore, elucidating the functional roles of lncRNAs in chronic liver disease and HCC can contribute to a better understanding of the molecular mechanisms and may help in developing novel diagnostic biomarkers and therapeutic targets for HCC as well as in preventing the progression of HCC from chronic liver disease [2]. In this review, some well-documented lncRNAs that participate in both hepatic fibrosis and HCC are comprehensively summarized.

## 2. The Classifications and Functions of lncRNAs

Numerous ncRNA molecules have been identified by RNA microarrays and the NGS of transcriptomes [13]. NcRNAs are important regulators involved in most cellular functions and modulation [14]. There have been three kinds of ncRNA classifications so far in the literature. First, they are classified into three types based on their relative sizes [11,15]. There are long ncRNAs (lncRNAs with more than 200 nucleotides), small ncRNAs (with less than 200 nucleotides) and microRNAs (miRNAs with 18–22 nucleotides) [15]. Second, lncRNAs are classified into five classes based on their genomic location relative to neighboring protein-coding genes: (1) sense lncRNAs overlap with the sense strand of a protein-coding gene; (2) antisense lncRNAs overlap one or more exons of a protein-coding gene on the opposite strand and initiate 3′ of a protein-coding gene; (3) bidirectional lncRNAs, in which the expression of a lncRNA and a protein-coding gene on the opposite strand are initiated <1000 base pairs away in close genomic proximity; (4) intronic lncRNAs are initiated completely within an intron of a protein-coding gene, without overlapping exons; and (5) intergenic lncRNAs (also termed large intervening ncRNAs or lincRNAs), are located near no other protein-coding loci [16,17,18]. Third, lncRNAs can also be classified according to their targeting mechanisms: signal, decoy, guide, and scaffold [3]. Increasing evidence has revealed that lncRNAs can interact with miRNAs. Indeed, lncRNAs can play a role of miRNA sponges (the so-called competing endogenous RNAs, ceRNAs), reducing their regulatory effect. In turn, miRNAs may directly interact with lncRNAs and silence their expression [19,20].

Many lncRNAs could not be easily classified into any particular category, and it is likely that the same lncRNAs may be listed in different groups in all classifications [21,22]. To date, the biological functions and molecular mechanisms of most lncRNAs remain largely unclear, with only a few being partially characterized [2]. Nevertheless, existing evidence demonstrates that these molecules play critical roles in regulating some cellular processes, specifically in the expression of protein-coding genes at the epigenetic, transcriptional and post-transcriptional levels [23,24,25]. Epigenetic alterations include changes in DNA methylation and histone modifications, as well as ncRNA-mediated gene silencing [26].

LncRNAs are considered to play regulatory roles in the pathogenesis and progression of various human diseases, including cell proliferation, differentiation, apoptosis, and tumorigenesis [27,28,29]. Many lncRNAs are frequently aberrantly expressed in human cancers in which they may act as oncogenes or tumor suppressors, indicating that they may serve as novel drivers of tumorigenesis [2,30]. Compared with protein-coding genes, the alterations of lncRNAs are highly tumor- and cell line-specific [31], and this characteristic of specificity makes lncRNAs promising diagnostic biomarkers [2].

Some studies suggested that HCC-related lncRNAs play critical regulatory roles in the development and progression of HCC, while their dysregulation is associated with diverse biological processes including proliferation, differentiation, apoptosis, invasion, and metastasis [22,32]. LncRNAs may also play a role in chemo-sensitivity or radio-resistance by blocking the cell cycle, suppressing apoptosis and reinforcing DNA injury repair [33]. For this reason, they can be used as potential targets for discovering new approaches to chemotherapy and radiotherapy in HCC-affected subjects, as demonstrated by Huang et al. [34].

## 3. LncRNAs Associated with both Chronic Liver Disease and HCC

Numerous lncRNAs that were increased in conjunction with inflammation and fibrosis were observed, and analyses of these transcripts found many pathways, including those involved in TGF-β1 and TNF signaling, ECM deposition, and insulin resistance [35]. LncRNAs found in animal models were also highly expressed in fibrosis-related NAFLD, including NEAT1, MALAT1, and PVT1. Another study suggested that some lncRNAs identified in a CCl4 mouse model, namely, APTR, MALAT1, NEAT1, and HOTAIR, may also be related to NAFLD fibrosis in humans [36].

In a study of hepatitis B virus (HBV) persistent carriers and HBV-positive HCC patients, Liu et al. [37] found two single nucleotide polymorphisms, rs7763881 in HULC and rs619586 in MALAT1. The data showed that mutations of both rs7763881 in HULC and rs619586 in MALAT1 were related to a decreased HCC risk. LncRNA levels were also increased in the patients with hepatitis C virus (HCV) infection [1]. HCV infection stimulates GAS5 and HOTAIR in HCV-infected liver cells [38,39]. Yuan et al. [40] reported that the plasma levels of LINC00152, RP11-160H22.5, and XLOC_014172 could effectively distinguish patients with HCC from chronic hepatitis patients and healthy controls. Another study also showed that serum levels of MALAT1 could differentiate HCC patients from HCV-induced liver cirrhosis patients and healthy controls [41].

An association of lncRNAs with metabolic disease has also been reported [1]. High expressions of lncRNAs in non-alcoholic steatohepatitis (NASH) patients showed that increased expressions of liver-specific lncRNA lnc18q22.2 were positively correlated with NASH grade and NAFLD score [42]. The deletion of LncLSTR, the liver-specific triglyceride regulator, reduced plasma triglyceride content by modulating metabolic pathways in mice [43]. In addition, mice with a genetic knock-out of lncRNA SRA, the steroid receptor RNA activator, were resistant to high fat diet-induced obesity, and SRA suppressed adipose triglyceride lipase, reducing free fatty acid β-oxidation and promoting hepatic steatosis [44].

An analysis of human lncRNAs from peripheral blood RNA identified that lncRNAs, such as AK128652 and AK054921, expressed in normal human plasma and liver, were significantly increased in patients with alcoholic cirrhosis [1]. These levels were inversely associated with the survival rate of patients with alcoholic cirrhosis [45].

As mentioned above, HCC-related lncRNAs play critical regulatory roles in the progression of HCC, while their dysregulation is related to diverse biological processes [32]. As a type of regulator of cellular processes including proliferation, apoptosis, and carcinogenesis, lncRNAs play important roles in the tumorigenesis and development of HCC [5]. An increased expression of lncRNAs in HCC is thought to have an oncogenic function, whereas lncRNAs exhibiting a decreased expression in HCC may act as tumor suppressors [2]. The dysregulation of HULC, HOTAIR, MALAT1 and other genes has been identified in HCC [5,46,47,48,49,50,51,52,53,54]. In the meta-analysis, high expression levels of 27 types of lncRNAs, such as AFAP-AS1, HOTTIP, and ZEB-1-AS1 were found to be related to a poor prognosis [55], and a low expression of 18 types of lncRNAs, such as GAS5, MEG3, and XIST, were found to be associated with a worse prognosis. LINC00052, ZEB1-AS1 and LINC01225 displayed oncogenic properties by facilitating the invasiveness and metastasis induction of HCC cells [39]. Some other lncRNAs, such as XIST, HOST2, HOXA-AS2, CCHE1, and AFAP1-AS1, can induce and accelerate cell proliferation, while inhibiting the apoptosis of HCC cells [33]. This feature might be useful as a therapeutic target of HCC, especially as an alternative to the patients who show chemoresistance for the chemotherapeutic drugs [6,7,8,56].

While the underlying mechanism of HCC-related lncRNAs remains largely elusive, understanding the differential expression and potential functional roles of lncRNAs in HCC, as well as the relationship in expression between chronic liver disease and HCC, is mandatory. Therefore, we searched manuscripts by the PubMed, using key words such as ‘lncRNA’, ‘liver’, ‘fibrosis’, and/or ‘HCC’. We found 32 lncRNAs associated with chronic liver diseases and/or HCCs. Among them, nine lncRNAs were selected which were associated with both fibrosis and carcinogenesis in the liver. Here, we briefly summarize nine well-documented lncRNAs that participate in both hepatic fibrosis and HCC. A few lncRNAs associated with hepatic fibrosis with unknown effects in HCC are also discussed. The detailed information on the lncRNAs described below is also summarized in Table 1.

### 3.1. HULC

Panzitt et al. [82] reported a ‘highly up-regulated in liver cancer’ (HULC) located on chromosome 6p24.3, as a novel ncRNA. HULC is implicated in hepatocarcinogenesis as an oncogene by regulating multiple biological processes [2]. HULC promotes the proliferation of HCC cells via the regulation of regulating cell cycle-related genes in HCC cells [59]. HULC is negatively associated with PTEN or miR-15a expression in HCC patients, which promotes malignant progression [83]. HULC contributes to malignant development by acting as a sponge of miRNAs such as miR-9, miR-107, and miR-372, which induce PPARα, E2F1, and cAMP response element-binding protein (CREB) respectively [47,48,49]. In addition, HULC is responsible for perturbations of the circadian rhythm by up-regulating the circadian oscillator CLOCK, clock circadian regulator, in HCC cells, leading to the promotion of hepatocarcinogenesis [84].

Zhao et al. [57] confirmed the effects of HULC on Tregs differentiation in HBV-related liver cirrhosis. They found that circulating Tregs and HULC were significantly up-regulated in HBV-related cirrhosis patients, and HULC regulates the function of Tregs by directly down-regulating the level of p18 [57]. An increased expression of HULC was also found in the liver tissue of high--fat-diet NAFLD rats. Shen et al. [58] investigated the role of HULC in hepatic fibrosis and hepatocyte apoptosis by inhibiting the MAPK signaling pathway in rats with NAFLD.

The levels of HULC were positively correlated with HBV X protein (HBx) in HCC patients [1]. The activation of HULC promotes HBx-mediated cell proliferation by inducing p18 [46]. Du et al. [46] reported that the up-regulation of HULC, mediated by HBx, promoted the proliferation of HCC through the down-regulation of the tumor suppressor gene, CDKN2C (p18).

HULC also induced epithelial mesenchymal transition (EMT), promoting tumor progression and metastasis by its competition with miR-200a [38,39]. In these studies, HULC expression was related to TNM stage, intrahepatic metastases, HCC recurrence, and postoperative survival [1,38,39]. Furthermore, HULC acts as a ceRNA to activate the EMT process through the HULC/miR-200a-3p/ZEB1 signaling pathway and stimulates HCC progression and metastasis [38,60]. Li et al. [85] demonstrates that HULC specifically binds to Y-box protein-1 (YB-1) to promote its phosphorylation through ERK kinase and then regulates the interaction of YB-1 with certain oncogenic mRNAs, consequently accelerating the translation of these oncogenic mRNAs in hepatic carcinogenesis.

### 3.2. MALAT1 (NEAT2)

Metastasis-associated lung adenocarcinoma transcript 1 (MALAT1), which is also known as nuclear-enriched abundant transcript 2 (NEAT2), is expressed in both human and mouse tissues and is located at chromosome 11q13 [2,5]. It is up-regulated in many malignant tumors and is involved in cell proliferation and migration via modulating caspase-3, caspase-8, Bax, Bcl-2, and Bcl-xL [51]. Aberrant MALAT1 expression promotes tumor metastasis by regulating gene expression and alternative pre-mRNA splicing [86,87].

In CCl4-treated mice, hepatic expression levels of MALAT1 were elevated in hepatic stellate cells (HSCs) and hepatocytes, respectively [62]. MALAT1 can promote the activation of HSCs by blocking the silent information regulator 1 (SIRT1)-induced inhibition of the TGF-β1 signaling pathway in the progression of liver fibrosis [11,61]. A recent study reported that hepatic expression of MALAT1 was higher in NASH patients than in those NAFLD patients with simple steatosis and in healthy controls [88]. It is also reported that MALAT1 acts as a ceRNA for miR-101b to regulate RAS-related C3 botulinum substrate 1 (Rac1), promoting proliferation, cell cycle progression, and HSC activation and contributing to hepatic fibrosis [62].

MALAT1 is up-regulated in HCC, and the overexpression of MALAT1 promotes cell proliferation, migration, and invasion in HCC. It is also correlated with the expression of HBx [2,5,89]. Furthermore, MALAT1 was identified to act as a highly sensitive marker of human HCCs, suggesting that MALAT1 can be used as a potential tool for HCC diagnosis [2,51]. Lai et al. [90] examined the role of MALAT1 in HCC prognosis. They observed that siRNA knockdown of MALAT1 reduced cell proliferation and repressed migration and invasion as well as apoptosis, indicating that blocking MALAT1 activity in HCC might be a vital anticancer therapy [90]. Huang et al. [91] showed the role of specificity protein (Sp) 1/3 in the transcriptional regulation of MALAT1 in HCC cells. They found that Sp1 and Sp3 are participated in the up-regulation of MALAT1 expression [91]. MALAT1 was identified to be up-regulated in HCC and to play a role of a proto-oncogene to enhance HCC cell growth via the activation of Wnt and mTOR pathway and the up-regulation of serine/arginine-rich splicing factor 1 (SRSF1) [2,63]. Therefore, the inhibition of SRSF1 expression or mTOR activity is able to block the oncogenic effects of MALAT1 and HCC development.

Given the presence of elevated MALAT1 levels in HCC, the biological functions of MALAT1 remain largely unclear and require further studies regarding their roles in the progression of chronic liver disease, from cirrhosis to HCC.

### 3.3. HOTAIR

Homeobox (HOX) transcript antisense intergenic RNA (HOTAIR) is a lncRNA that resides on a boundary of the HOXC locus on chromosome 12q13.13, which is co-expressed with HOXC genes [2,92,93]. An increasing number of studies have investigated whether HOTAIR is up-regulated in multiple cancers, including breast cancer, lung adenocarcinoma, renal cell carcinoma, pancreatic cancer, and HCC [94,95,96,97,98].

Yang et al. [98] reported that the expression of HOTAIR in HCC tissues is significantly increased, when compared to that in adjacent non-cancerous tissues. In addition, the levels of HOTAIR expression in HCC cell lines were elevated compared to those in normal liver cell lines [98]. Notably, HCC patients with a high expression of HOTAIR had significantly worse prognoses than those without expression of HOTAIR [50]. HCC patients with HOTAIR overexpression have an increased risk of recurrence after hepatectomy, and HOTAIR overexpression is also correlated with increased risk of lymph node metastasis [99]. Furthermore, patients with a high expression of HOTAIR have a significantly shorter recurrence--free survival than patients with a low expression of HOTAIR [100].

To understand the oncogenic functions of HOTAIR in HCC, various mechanisms have been suggested [2]. The up-regulation of HOTAIR induces the proliferation, migration, and invasion of human HCC cells through the activation of autophagy [51,53,54]. Ding et al. [53] suggested that HOTAIR plays a critical role in the progression of HCC via inhibition of RNA binding motif protein 38 (RBM38). A regulatory network between miR-218 and HOTAIR was identified, whereby HOTAIR inactivates P16 (Ink4a) and P14 (ARF) signaling by down-regulating the expression of miR-218, resulting in hepatocarcinogenesis [52].

HOTAIR expression was up-regulated in the livers of CCl4-treated mice [64]. The expression levels of HOTAIR were also increased in cirrhotic liver tissues from HBV patients and colocalized with ACTA2, indicating that HSCs may be the primary source for HOTAIR in fibrotic liver [36]. A functional characterization of the lncRNA demonstrated that the overexpression of HOTAIR induced cell proliferation and elevated levels of ACTA2 and COL1A1, as well as fibrosis-related genes, such as matrix metalloproteinase 2 (MMP2) and MMP9 [64]. In addition, HOTAIR functions as a ceRNA to sponge miR-29b and then represses DNA methyltransferase 3b (DNMT3b), resulting in up-regulation of PTEN methylation, which contributes to hepatic fibrosis [65]. These results suggest that HOTAIR may promote fibrosis in liver by regulating DNMT1, MEG3, and the p53 pathway in HSCs, although further studies on this lncRNA is necessary in order to identify the role of HOTAIR in the process of hepatic fibrogenesis and carcinogenesis.

### 3.4. MEG3

Maternally expressed gene 3 (MEG3), is a lncRNA located at human chromosome 14q32 [101]. MEG3 could activate p53 to induce caspase-3-dependent apoptosis and suppress the expression of col1α1 and α-smooth muscle actin (α-SMA) in activated HSCs [66]. It is suggested that MEG3 plays a critical role in the activation of HSCs and fibrogenesis in liver [11].

One of the first studies of lncRNAs in NAFLD fibrosis identified that the expression of MEG3 was decreased in the livers of CCl4-treated mice, when compared to those of oil-fed control animals, and that the expression of MEG3 reduced concordantly with the progression of fibrosis [66]. In contrast to these results, hepatic MEG3 levels were significantly increased in liver fibrosis and NASH cirrhosis in human patients [102].

Numerous investigations have evaluated the functional role of MEG3 as a tumor suppressor in various types of human cancers, such as gastric cancer, lung cancer, glioma, cervical cancer, bladder cancer, and HCC [103,104,105,106,107,108,109]. The expression of MEG3 was found to be markedly decreased in HCC tissues [67,110]. Furthermore, ectopic MEG3 expression in HCC cells significantly inhibits proliferation and mediates apoptosis [67,105,110]. Several studies have also determined that MEG3 overexpression results in an increase of p53 protein and stimulates its transcriptional activity in HCC cells [67,111].

MEG3 is suggested to be an independent prognostic factor for HCC because the low expression of MEG3 was associated with a worse prognosis, compared to the high expression of MEG3 in HCC patients [111]. The biological roles of MEG3 also remain largely unknown and require further investigation regarding its function in the progression of chronic liver disease to HCC.

### 3.5. lncRNAp21

The long intergenic non-coding RNA-p21 (lncRNAp21), which is located 15 Kb upstream of the gene encoding the critical cell cycle regulator Cdkn1a (also known as p21), contains two exons comprising 3.1 Kb, together [112]. LncRNA-p21 acts as a transcriptional suppressor in the p53 pathway by activating p53 to promote apoptosis [113]. It has been investigated that lncRNAp21 is deregulated in various human diseases, such as skin tumors, prostatic cancers, and HCCs [114,115,116]. It also functions as a tumor suppressor in malignancies, but the mechanism of the process remains elusive [115].

Serum levels of lncRNA-p21 were negatively correlated with liver fibrosis in HBV patients [117]. Yu et al. [68] determined that lncRNA-p21 enhanced the expression of PTEN by sequestering miR-181b as a ceRNA and inhibited HSC activation through the PTEN/Akt pathway in liver fibrosis [68]. It was also found that the lncRNA-p21 competitively binds miR-17-5p for the inhibition of WIF1 via the Wnt/β-catenin pathway leading to the suppression of HSC activation [69].

### 3.6. GAS5

Growth arrest-specific transcript (GAS) 5 was initially discovered in a screen for potential tumor suppressor genes that are highly expressed during growth arrest. GAS5 was originally isolated from mouse embryo NIH 3T3 cells [118].

GAS5 has been reported as a tumor suppressor in some cancers, and it is related to the proliferation, apoptosis, and migration of tumor cells in breast cancer, stomach cancer and prostate cancer [119,120,121]. GAS5 directly binds to miR-21 to down-regulate its expression and negatively regulate the expression of miR-21 in HCC [71]. In liver fibrosis, GAS5 interacts with miR-222 and promotes the expression of p27 protein, thereby inhibiting the activation and proliferation of HSCs [70].

### 3.7. PVT1

Plasmacytoma variant translocation 1 (PVT1) is transcribed from a locus adjacent to the MYC locus on human chromosome 8q24 [72]. The PVT1 is known to be up-regulated in some human tumors, such as HCC, ovarian cancer, malignant pleural mesothelioma, non-small cell lung cancer and renal cancer [122,123,124,125,126].

The silencing of PVT1 in primary HSCs not only reduced the cell proliferation, but also decreased the protein levels of Actα2 and Col1α1 [36]. The PVT1 knockdown in primary HSCs was also associated with changes in the markers of the EMT process, indicating a possible mechanism by which this lncRNA promotes hepatic fibrosis [127].

PVT1 also activates the Hedgehog pathway by enhancing the methylation of Patched1 (PTCH1) and down-regulating PTCH1 expression through competitively binding miR-152, which is a driver of EMT and HSC activation in hepatic fibrosis [72].

### 3.8. NEAT1

Nuclear-enriched abundant transcript 1 (NEAT1) is located in 11q13 [128]. The expression of NEAT1 was found to be increased in primary HSCs derived from CCl4-treated mice compared to control mice [74]. NEAT1 overexpression promoted the activation of HSCs and increased levels of Actα2 and Col1α1, indicating that this lncRNA plays a role in HSC activation [36]. NEAT1 overexpression corresponded with decreased levels of miR-122, which was identified to regulate NEAT1 effects on the activation of HSCs by a mechanism associated with Kruppel-like factor 6 (Klf6) [36,74]. The expression of NEAT1 was up-regulated in HCC, while its knockdown was correlated with decreases not only in HCC cell proliferation, but with also invasion, and migration via regulating heterogeneous nuclear ribonucleoprotein A2 (hnRNP A2) [75]. However, the role of NEAT1 expression in cancer is controversial, because NEAT1 is strongly up-regulated by p53 in several cancers, and appears to play a tumor-suppressive role in the presence of wild-type p53 activation [129,130,131]. To understand the molecular mechanism of NEAT1 in chronic liver disease and HCC properly, the effect of p53 activation and mutation status always needs to be considered [131].

### 3.9. LncRNA-ATB

Qiu et al. [132] propose that lncRNA-ATB, activated by TGF-β1, could be used as a novel diagnostic biomarker to identify the severity of inflammation and fibrosis. Some studies investigated lncRNA-ATB, a lncRNA associated with liver fibrosis and HCC, in HCV patients and identified that plasma levels of this lncRNA were significantly correlated with the stage of liver fibrosis [133,134]. LncRNA-ATB was also significantly increased in HCC tissues and positively correlated with intrahepatic or extrahepatic metastases [135].

### 3.10. Other lncRNAs Related to Liver Fibrosis

The liver fibrosis-associated lncRNA1 (lnc-LFAR1) is a 734 nt transcript, which was originally demonstrated as a liver-enriched lncRNA in the fibrotic livers of mice [11]. Zhang et al. [77] used microarray analysis to profile lncRNAs in CCl4-treated mice and identified 266 up-regulated and 447 down-regulated lncRNAs [77]. Of these, a single lncRNA was identified, and it was most abundantly expressed in hepatocytes, followed by HSCs and Kupffer cells [77]. The lnc-LFAR1 of mice is located in chromosome 4q25, and it is adjacent to the CYP2U1 and HADH genes, which also exist in humans [77]. A recent analysis showed that three potential Smad2/3 binding sites (SBE) are present in the promoter of lnc-LFAR1, which means that Smad2/3 can bind to the promoter of lnc-LFAR1 to increase its expression [11,77]. Lnc-LFAR1 binds directly to Smad2/3 to control the transcription of a number of genes, including TGFB1, PAI, ACTA2, COL1A1, Smad2, Smad3, Notch2, and Notch3, resulting in the activation of the TGF-β1 and Notch pathways [36,77]. Together, lnc-LFAR1 was found to exert effects on HSC activation, hepatocyte apoptosis, and hepatic fibrogenesis in a mouse model [36,77].

Negishi et al. [136] identified a novel lncRNA, the Alu-mediated p21 transcriptional regulator (APTR), which was demonstrated to modulate cell cycle progression and cell proliferation. In an independent study, the expression of APTR was observed to be more than twofold higher in the fibrotic livers of animal models and cirrhotic livers in humans [79]. The silencing of APTR in primary HSCs also reduced ACTA2 and COL1A1 mRNA and protein expression and suppressed the TGF-β1-mediated up-regulation of ACTA2 [36]. In individuals with a cirrhotic liver, serum levels of APTR were approximately four-fold higher than individuals with a normal histology, and two-fold higher in patients with decompensated cirrhosis than those with compensated cirrhosis [79], indicating that serum levels of APTR may also serve as biomarkers of the severity for liver fibrosis [36].

LncRNA HIF1A-AS1 interacts with the partner TET3, one member of the ten-eleven translocation (TET) protein family, which is associated with DNA methylation, to inhibit the HSC activation [11,78]. The levels of both Cox2 and lncRNA-Cox2 were elevated in CCl4-treated mice, when compared to control animals, and those two transcripts were positively correlated with the amount of tissue affected by fibrosis [80,81].

## 4. Conclusions

It is believed that numerous lncRNAs are correlated with the progression of both liver cirrhosis and HCC from a review of the literature. Therefore, we can suppose that such lncRNAs play a crosslinking role between chronic liver disease and HCC. With the rapid development of high technology, such as NGS and bioinformatics, an increasing number of lncRNAs are emerging as novel biomarkers for early diagnosis, better prognostic evaluation and efficient therapeutic targets for HCC in future clinical applications [5]. It has been demonstrated that LncRNAs found in body fluids can be used as fluid-based non-invasive biological markers for clinical trials. In addition, lncRNAs can influence the sensitivity of HCC to chemo- or radiation therapy [5]. Thus, in the future, a better understanding of the molecular mechanism of lncRNAs associated with the initiation and progression of HCC will provide a rationale for novel effective lncRNA-based targeted therapies. Genomic/epigenetic directed stratifications in clinical trial design and enrollments in the era of lncRNAs could be considered. One of the underlying messages is that more precision and more individualized approaches need to be tested or considered in well-designed clinical trials [137,138]. Further studies are required, in particular, to examine the toxicity and the pharmacokinetics of lncRNAs and, additionally, to evaluate their biological properties for chronic liver disease. Nevertheless, these lncRNAs have the potential to be novel promising tools for HCC diagnosis and prognosis, as well as the protection of HCC development from chronic liver disease.

## Figures and Tables

**Table 1 ijms-21-02883-t001:** Long non-coding RNAs (lncRNAs) associated with both chronic liver disease and hepatocellular carcinoma (HCC).

LncRNAs	Location	Dys-Regulation	Associated Hepatitis	In Liver Fibrosis		In Hepatocellular Carcinoma		References
				Function	Mechanism	Function	Mechanism	
HULC	6p24.3	Up	hepatitis B	-Treg differentiation -hepatocyte apoptosis	-p18 downregulation -inhibit MAPK pathway	oncogene; proliferation, EMT	- regulate cell cycle-related genes - miR-200a-3p/ZEB1 pathway	[38,46,47,48,57,58,59,60]
MALAT1	11q13	Up	CCl4-Tx mice NASH hepatitis B	HSC activation	activate TGF-β pathway and Rac1	oncogene	- modulate apoptosis - mTOR and Wnt pathway	[51,61,62,63]
HOTAIR	12q13.13	Up	CCl4-Tx mice hepatitis B	inhibit apoptosis express fibrosis-related genes	- DNMT1-MEG3-p53 pathway - PTEN methylation	oncogene	- repress RBM38 - inactivate p16 (Ink4a) and p14 (ARF)	[51,52,53,54,64,65]
MEG3	14q32	Down ^1^	CCl4-Tx mice	apoptosis	activate p53	tumor suppressor; activate apoptosis	p53 target gene expression	[66,67]
LncRNAp21	6p21	Down	hepatitis B	Suppress HSC activation	-enhance PTEN/Akt pathway -inhibit Wnt/β -catenin pathway	tumor suppressor	Unclear	[68,69]
GAS5	1q25	Down	primary HSC	inhibit HSC activation and proliferation	promote p27 expression	tumor suppressor	bind to miR-21	[70,71]
PVT1	8q24	Up	primary HSC	EMT and HSC activation	hedgehog pathway	oncogene	- recruit EZH2, - inhibit P53 expression	[72,73]
NEAT1	11q13	Up ^1^	CCl4-Tx mice	HSC activation	Neat1-miR-122-Klf6 axis	proliferation and migration	regulate hnRNP A2	[74,75]
LncRNA-ATB	14	Up	hepatitis C	HSC activation	regulate TGF-β pathway	autophagy	YAP and ATG5 expression	[76]
Lnc-LFAR1	4q25 ^2^	Up	CCl4-Tx mice	HSC activation, hepatocyte apoptosis	TGF-β and Notch pathway	unknown	not established	[77]
HIF1A-AS1	14q23.2	Down	primary HSC	HSC inactivation	interact with TET3	unknown	not established	[78]
APTR	7q11.23	Up	CCl4 and BDL Mice ^3^	HSC activation	TGF-β pathway	unknown	not established	[79]
LncRNA-Cox2	1q25	Up	CCl4-Tx mice	unknown	not established	unknown	not established	[80,81]

^1^ Controversial, ^2^ mouse chromosome, ^3^ including undisclosed etiology of human liver fibrosis; ATG5; autophagy-related gene 5, BDL; bile duct ligation, CCl4-Tx; CCl4-treated, DNMT1; DNA methyltransferase 1, EMT; epithelial mesenchymal transition, EZH2; enhancer of zeste homolog 2, hnRNP A2; heterogeneous nuclear ribonucleoprotein A2, HSC; hepatic stellate cell, Klf6; Kruppel-like factor, RBM38; RNA binding motif protein 38, TET3; ten-eleven translocation 3, YAP; yes-associated protein.

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
