# Peer review of "LncRNAs Act as a Link between Chronic Liver Disease and Hepatocellular Carcinoma"

_ijms, 2020, doi:10.3390/ijms21082883_

Round 1

Reviewer 1 Report

Comments to the Authors

The authors submitted an interesting review regarding long non-coding RNAs (lncRNAs), focusing their attention on the emerging role of these molecules in pathogenesis of chronic liver disease and hepatocellular carcinoma.

This manuscript was well written and I believe with small adjustments could add value to the literature.

Minor revisions:

1) Line 55 The classification relative to the size is referred to ncRNA, so the authors should substitute “lncRNA” with “ncRNA”.

2) Line 58 The authors should substitute “they” with “LncRNA” are classified…

3) Line 62 and 327 The authors should substitute “an lncRNA” with “a lncRNA”

4) Line 98 The authors should substitute “studies” with “study”

5) Line 220 The authors should substitute “compare” with “compared”

Author Response

Dear. Reviewer 1,

Thank you for your kind comments.

As your suggestion, we substituted all of the mentioned words with your recommendations and highlighted the changes in the revised manuscript.

1) Line 55 substitute “lncRNA” with “ncRNA”.

2) Line 58 substitute “they” with “LncRNA” are classified…

3) Line 62 and 327 substitute “an lncRNA” with “a lncRNA”

4) Line 98 substitute “studies” with “study”

5) Line 220 substitute “compare” with “compared”

Reviewer 2 Report

This review is well described and seems to be beneficial for researchers involved in this area. However, some points should be improved.

In the section of NEAT1, the authors mainly introduce oncogenic roles of NEAT1. However, it is revealed that NEAT1 is strongly upregulated by p53. (Adriaens, et al. Nat Med 2016, Idogawa, et al. Int J Cancer 2017) and the effect of p53 activation and mutation status always need to be considered in any pathways and diseases (Idogawa, et al. J Oncol 2019). The authors should cite these papers in the main text and review this point adequately.

Author Response

Dear Reviewer 2,

We are grateful to you for the critical comment and useful suggestion that have helped us to improve our paper considerably.

Comment: In the section of NEAT1, the authors mainly introduce oncogenic roles of NEAT1. However, it is revealed that NEAT1 is strongly upregulated by p53. (Adriaens, et al. Nat Med 2016, Idogawa, et al. Int J Cancer 2017) and the effect of p53 activation and mutation status always need to be considered in any pathways and diseases (Idogawa, et al. J Oncol 2019). The authors should cite these papers in the main text and review this point adequately.

Answer: Thank for your precise suggestion for our manuscript. As your suggestion, we added some sentences with the citations as below in the section of NEAT1 and highlighted the changes in the revised manuscript.

Added sentences : However, the role of NEAT1 expression in cancer is controversial, because NEAT1 is strongly up-regulated by p53 in several cancers, and appears to play a tumor-suppressive role in the presence of wild-type p53 activation[129-131]. To understand the molecular mechanism of NEAT1 in chronic liver disease and HCC properly, the effect of p53 activation and mutation status always need to be considered [131].

Reviewer 3 Report

To the authors

Young-Ah Kim and colleagues provide a balanced assessment of the status of lncRNAs in HCC and chronic liver diseases such as NAFLD and fibrosis. The article highlights important data that might have been overlooked when promulgating the clinical value of epigenetic modification and gene-expression that controlled the progression of HCC. The author findings provided potential targets for prevention and treatment of live cancer-related trials. Nonetheless, a few points should be considered in order to enhance the scientific soundness and audience resonance of the topic discussed.

General methodological comment: how the author’s literature search and subsequent methodological analysis of accumulated papers were done, remains unexplained and would need at least a comment, if possible, an illustrative Figure and/or description of the Medline/manuscript searching criteria utilized.

Clinical outlook and therapeutic window: Despite very exhausting, I personally miss some recent work’s aspects, fitting with the well-described biological It has been described that LncRNAs can impact on fibrosis: remarkably, the authors themselves point out for instance that “HULC acts as a ceRNA to activate the EMT process through the HULC/miR-200a-3p/ZEB1 signalling pathway and stimulates HCC progression and metastasis”; fibrosis and oncogenic signalling relationship within the HCC physiopathology is as relevant as a description of LncRNAa, especially by considering a truncal consequence of biological dissection, namely drug resistance (PMID: 31640191; PMID: 30105249): “shoe-horn in” these concepts into this manuscript would open up a transversal pathophysiological principle in the context of epigenetics and sorafenib/TKI resistance in fueling neoplastic cells. Specifically, both the partially neglected miRNA-21 and TUC338 play an emerging role in drug-resistant phenotype and pathological fibrosis (PMID: 31640191; PMID: 26131450)

In the frame of the above-mentioned thinking, the authors could provide a little more consideration of genomic/epigenetic directed stratifications in clinical trial design and enrollments. One of the underlying messages here is that more precision and individualized approaches (taking into account can also increase the potential to open a theranostic window on HCC milieu too as a non-bystander rather active driver of malignancy) need to be tested or considered in well-designed clinical trials – a challenge, but I would be interested in their perspective of how this might be done, also in light of recently published manuscript (PMID: 32083005; PMID: 31627433).

The manuscript would benefit from linguistic editing.

Author Response

Dear Reviewer 3,

We are grateful to you for the critical comments and useful suggestions that have helped us to improve our paper considerably.

All modifications are highlighted in the revised manuscript.

Comment 1. General methodological comment: how the author’s literature search and subsequent methodological analysis of accumulated papers were done, remains unexplained and would need at least a comment, if possible, an illustrative Figure and/or description of the Medline/manuscript searching criteria utilized.

Answer: I understand why your comments are important.

First of all, we searched manuscripts by the PubMed, using key words such as lncRNA, liver, fibrosis, and/or HCC. Then, we could find 32 lncRNAs associated with chronic liver diseases and/or HCCs. Among them, nine lncRNAs were selected which were relating to both fibrosis and carcinogenesis in liver. We added these sentences in last paragraph of section 3 (LncRNAs associated with both chronic liver disease and HCC) right in front of Table 1 and highlighted the changes in the revised manuscript.

Comment 2. Clinical outlook and therapeutic window: Despite very exhausting, I personally miss some recent work’s aspects, fitting with the well-described biological It has been described that LncRNAs can impact on fibrosis: remarkably, the authors themselves point out for instance that “HULC acts as a ceRNA to activate the EMT process through the HULC/miR-200a-3p/ZEB1 signalling pathway and stimulates HCC progression and metastasis”; fibrosis and oncogenic signalling relationship within the HCC physiopathology is as relevant as a description of LncRNAa, especially by considering a truncal consequence of biological dissection, namely drug resistance (PMID: 31640191; PMID: 30105249): “shoe-horn in” these concepts into this manuscript would open up a transversal pathophysiological principle in the context of epigenetics and sorafenib/TKI resistance in fueling neoplastic cells. Specifically, both the partially neglected miRNA-21 and TUC338 play an emerging role in drug-resistant phenotype and pathological fibrosis (PMID: 31640191; PMID: 26131450)

Answer: Thank you for your comment. As your suggestion, we revised the sentences of introduction part and section 3 (LncRNAs associated with both chronic liver disease and HCC) as below, added citations, and highlighted the changes in the revised manuscript.

  1. Introduction part

Original sentence: Despite advances in the understanding of the molecular mechanisms of HCC, the overall low survival time, high rates of metastasis and postsurgical recurrence are still unsolved problems [5].

Revised sentence: Despite advances in the understanding of the molecular mechanisms of HCC, the overall low survival time, high rates of metastasis, postsurgical recurrence, and chemoresistance are still unsolved problems [5-8].

  1. Section 3. LncRNAs associated with both chronic liver disease and HCC

Original sentence: This feature might be useful as a therapeutic target of HCC [53].

Revised sentence: This feature might be useful as a therapeutic target of HCC, especially as an alternative to the patients who show chemoresistance for the chemotherapeutic drugs [6-8,56].

Comment 3. In the frame of the above-mentioned thinking, the authors could provide a little more consideration of genomic/epigenetic directed stratifications in clinical trial design and enrollments. One of the underlying messages here is that more precision and individualized approaches (taking into account can also increase the potential to open a theranostic window on HCC milieu too as a non-bystander rather active driver of malignancy) need to be tested or considered in well-designed clinical trials – a challenge, but I would be interested in their perspective of how this might be done, also in light of recently published manuscript (PMID: 32083005; PMID: 31627433).

Answer: Thank for your precise suggestion for our manuscript. As your suggestion, we added some sentences and citations in conclusion part as below and highlighted.

Added sentences: It could be considered the genomic/epigenetic directed stratifications in clinical trial design and enrollments in era of lncRNAs. One of the underlying messages would be more precision and individualized approaches need to be tested or considered in well-designed clinical trials [137,138].

Round 2

Reviewer 2 Report

The paper has been revised adequately. 

Reviewer 3 Report

The authors have clarified several of the questions I raised in my previous review. Most of the major problems have been addressed by this revision. No more comment from this reviewer.